# Patterns of Online Stress Management Information-Seeking Behavior in Hungary

**DOI:** 10.3390/ijerph22040473

**Published:** 2025-03-22

**Authors:** György Jóna, Anita R. Fedor

**Affiliations:** Faculty of Health Sciences, University of Debrecen, Sóstói út 2-4, 4400 Nyíregyháza, Hungary; fedor.anita@etk.unideb.hu

**Keywords:** online stress management information-seeking (OSMIS) behavior, digital inclusion, health inequalities, techquity

## Abstract

This paper examines the societal, demographic, and health-related determinants of online stress management information-seeking (OSMIS) behavior in Hungary. We processed the International Social Survey Program: Health and Healthcare (n = 1008) dataset of 2021. Relationships between variables were assessed using weighted multiple logistic regression. The bootstrapping method was applied to gauge the robustness and reliability of the estimates. Subgroup analyses were also utilized to explore potential confounding effects between OSMIS behavior and various socioeconomic and health-related lifestyle factors. Empirical findings indicate that socially excluded strata were the most likely to seek online stress management information to cope with stressful situations. OSMIS behavior was significantly associated with divorced marital status (OR = 3.13; 95% CI: [1.92–5.17]), unemployment (OR = 2.22 [1.64–2.99]), living in a rural village (OR = 1.39 [1.12–1.93]), and distrust in the healthcare system (OR = 2.03 [1.33–3.11]). During the COVID-19 pandemic, the concept of techquity played a pivotal role in Hungary, bridging gaps in health access. Policymakers, healthcare practitioners, and digital health developers may harness our results to enhance digital health tools within integrated healthcare systems, prioritizing equitable access to ensure that marginalized populations can fully benefit from the advantages of techquity and digital inclusion.

## 1. Introduction

Stress significantly determines the prevalence of morbidity, and the principle causes of death, such as cardiovascular diseases, different types of cancers, dementia, and immune system suppression. In addition, stress profoundly influences human behavior, often manifesting as anxiety, depression, fear, aggression, sadness, or frustration. The remarkable rate of mental diseases and morbidity is correlated with stress and its deteriorative effects cannot thus be overestimated [1,2,3]. Since detrimental symptoms of stress affect all members of modern societies, it must thus be learned how to cope with it effectively.

Nowadays, individuals collect diverse information from the internet; this learning process of stress management might be supported by some websites demonstrating techniques and special methods of stress relief [4]. Connected devices, for example smartphones, laptops, tablets, desktop computers, play a central role in easily accessing online health information on coping strategies and how to internalize them [5,6]. The usage of connected devices increased globally, which has even been further accelerated during the COVID-19 pandemic [7,8,9]. More precisely, the usage of digital tools for health-related information-seeking varies across regions, with Finland having the highest proportion (77%) of individuals searching for health information online, followed by the Netherlands (76%), Denmark (72%), and Germany (70%). In contrast, lower percentages were observed in Romania (28%), Bulgaria (29%), and Italy (35% in 2019) [10,11]. Nevertheless, a relatively high rate of the Hungarian population has access to the internet (2020: 87.6%, 2021: 90.8%, 2022: 91.4%, 2023: 92.7%) due to its low cost [12].

In principle, online health information should be distributed across countries, allowing socially excluded groups to access a variety of health services, thereby contributing to the reduction in health inequalities. This process is encapsulated in the concept of ‘techquity’, a relatively new paradigm in digital health studies. The term ’techquity’ combines ’technology’ and ’equity’, highlighting the efforts to ensure equitable access to digital health resources across different social strata [13].

Empirical studies to date have investigated the extent to which individuals seek comprehensive information on health, health-related lifestyle, mental well-being, and prevention [14,15,16]. Notwithstanding, the socioeconomic and demographic determinants of online stress management information-seeking (OSMIS) behavior remained underexplored. This paper addresses this gap.

OSMIS behavior has the potential to reshape stress management patterns at the population level. OSMIS is defined as the proactive online search for stress management information, in which individuals collect, evaluate, and internalize different stress management techniques [16]. Through connected devices, individuals can access a wide range of affordable, practical, and accessible coping strategies, including positive thinking techniques, art or music therapy, time management tips, nutritional guidance, and other psychological and mental health advice from healthcare professionals [17,18,19]. Since these platforms typically offer online stress management information (OSMI) at no cost, they enable individuals from diverse socioeconomic backgrounds to access online stress management resources without financial or geographic barriers. As such, OSMIS behavior has the potential to reduce health inequalities while simultaneously improving mental well-being and resilience [20]. The convenience, anonymity, and user-friendly nature of these platforms further promote OSMIS behavior, particularly for individuals who may avoid traditional forms of help due to stigma, financial concerns, or privacy issues [21,22].

The subsequent chapters of this paper explore the Hungarian socioeconomic, health-related lifestyle, and demographic determinants of OSMIS behavior.

## 2. Materials and Methods

### 2.1. Study Design

We used the Hungarian cross-sectional, primary database from the 2021 iteration of the International Social Survey Program: Health and Healthcare (ISSP 2021 dataset). Data collection was conducted from April to May 2021 during the COVID-19 pandemic. We processed this primary database in this study. It embraced a comprehensive array of health data, including subjective health perception conditions, life satisfaction, happiness, the prevalence of various diseases, confidence in healthcare systems, various demographic indicators, and the status of social support as well. In brief, this extensive dataset provided a rich foundation for scrutinizing the statistical relationship between various socioeconomic variables and OSMIS behavior metrics.

### 2.2. Questionnaire, Sample Population, and Data Acquisition

The applied questionnaire was developed as follows. In 1985, a Basic Questionnaire was established, encompassing a set of background variables (e.g., sex, age, and income). In addition to the mandatory module questions, preselected optional questions and country-specific variables can be included. The present standard and an overview of past modifications are recorded in the Background Variable Guidelines, which offer comprehensive details on measurement objectives, coding structures, filtering criteria, and potential question formats. Essentially, the questionnaire consists of comprehensive, standardized questions (Basic Questionnaire), which are implemented in every participating country, as well as country-specific variables. The Hungarian and English questionnaires can be downloaded from here Appendix A. The translation of this questionnaire adhered to strict procedures and scientific guidelines to ensure the accuracy, validity, and reliability of the instrument across diverse linguistic and cultural contexts. The English version was translated into Hungarian using the method ‘Forward and Backward Translation Procedure’. The standardized questionnaire was administered using electronic and in-person (paper-and-pencil: PAPI) methods, with assistance from professional interviewers. The ISSP 2021 dataset, curated and overseen by the ISSP and the TÁRKI Social Research Institute (Hungary), was collected through a meticulous multistage random sampling method, ensuring its relevance to both young adult and older adult populations in Hungary. The large Hungarian sample comprised 1008 respondents. Individualized weighting was employed to reduce nonresponse bias and maintain the representativeness of the sample, in accordance with ISSP guidelines. Details of the survey’s sampling frame, original questionnaires, and methodology can be found at https://www.gesis.org/en/allbus/data-and-documentation (accessed on accessed on 11 March 2025). All data used in this study were fully anonymized prior to access. This study was carried out in accordance with the guidelines and principles of the Declaration of Helsinki.

### 2.3. Data Treatment and Variable Specification

In this survey, the selection of applied variables was based on a meticulous and careful review of the scientific literature on similar topics. OSMIS behavior, as a dependent variable, was measured using Question 12/B: ‘During the past 12 months, how often, if at all, have you used the internet to look for information on the following topics? Information related to stress’. Response options included never, seldom, sometimes, often, and very often. We recoded these into a binary variable measuring OSMIS behavior, where 0 = never or seldom, and 1 = sometimes, often, or very often searching for online information about stress management. Sex was delineated into male and female. Age was classified into four groups: 18–24 years (Generation Z), 25–43 years (Generation Y), 44–59 years (Generation X), and over 60 years old (Older Adults). Marital status was stratified into single, married, divorced, and widowed. Educational attainment was categorized into three levels: primary, secondary, and higher education. Employment status was divided into three groups: employed, unemployed, and pensioner. The place of residence was coded as big city, suburb, town/small city, or rural village.

Economic background was examined using two approaches: a subjective self-assessment categorized as poor, lower middle class, middle class, upper middle class, or affluent, and an objective classification into quintiles, ranging from the first quintile (marginalized social class) to the fifth quintile (élite). Healthy lifestyle variables were also incorporated into this study. Smoking and alcohol consumption were dichotomized into smoker vs. non-smoker and drinker vs. non-drinker, respectively. Depression and chronic disease were also classified as either yes or no. Happiness and self-perceived health were dichotomized as either not good or good, and not happy or happy, respectively. Body mass index (BMI) was categorized as overweight/obese or normal. Religion was coded as non-religious, Catholic, and other religions. Attendance at religious services was categorized as several times a month, several times a year, or never. Trust in healthcare, physicians, and online health information was dichotomized as either yes or no. Finally, respondents could answer whether access to healthcare was easy with yes or no.

### 2.4. Statistical Analysis

Descriptive statistical analyses were conducted by applying weighted proportions and *p*-values to assess the significance of associations between variables, which guided the selection of variables for the regression model. Following the finalization of the predictor set, a weighted multiple logistic regression model was constructed [23]. The threshold for statistical significance was established at a p-value of less than 0.05. The outcomes of the logistic regression analysis were presented as adjusted odds ratios (ORs) with 95% confidence intervals (CIs). The multiple logistic regression model was formed with the stepwise variable selection method. All statistical analyses were conducted employing IBM SPSS Version 30.0 software.

### 2.5. Sensitivity Analysis

Bootstrapping methods were used to evaluate the robustness and reliability of the logistic regression estimates. Specifically, 1000 bootstrapped samples were generated through resampling with replacement, and these samples were subsequently analyzed within the context of the original logistic regression model [24,25]. Additionally, subgroup analyses were conducted to investigate potential confounding effects on the relationship between OSMIS behavior indicators and various socioeconomic, healthy lifestyle, and demographic variables. These methods aimed to assess the consistency of the observed associations across the different demographic subgroups identified in the primary analysis [26].

## 3. Results

### 3.1. Socioeconomic, Demographic, and Healthy Lifestyle Characteristics

Sex emerged as a key demographic factor, with the highest prevalence of OSMIS observed among females (69.1%, *p* < 0.001). The rate of OSMIS behavior was significantly higher among Generation Y and Z compared to older adults (37.7%, *p* < 0.001; 36%, *p* < 0.001, respectively). Marital status also displayed a strong association with OSMIS, with the highest prevalence found among married individuals (58.3%, *p* < 0.001). Moreover, the proportion of OSMIS was the highest among secondary or higher education individuals (77.2%, *p* < 0.001).

Employment status proved to be a significant determinant, as employed individuals were more likely to seek OSMI compared to older adults (75.6%, *p* < 0.001). The prevalence of OSMIS was particularly elevated in the second income quintile (43.2%, *p* < 0.001). A stark contrast emerged with respect to alcohol consumption, where non-drinkers exhibited the highest rate of OSMIS (94.6%, *p* < 0.001). Furthermore, not having chronic illnesses showed a significantly higher prevalence of OSMIS (81.8%, *p* < 0.001). Happiness levels were also strongly correlated with OSMIS, as individuals who rated their happiness as good displayed a prevalence of 67.8% (*p* < 0.001). The prevalence of OSMIS was significantly higher among normal BMI compared to respondents who were overweight or obese (74.4%, *p* < 0.04). A significant relationship was observed between OSMIS and self-perceived health status, with individuals perceiving their health as good showing a rate of 46.2% (*p* < 0.001).

Notably, the prevalence of OSMIS was high among Roman Catholic individuals (47.8%, *p* < 0.001). The proportion of OSMIS was substantially higher among those who reported a higher level of trust in online health information and healthcare schemes (90.95%, *p* < 0.001; 91.5%, *p* < 0.001, respectively). Finally, the rate of OSMIS was high among individuals who perceived the healthcare system as easy to access (74.2%, *p* < 0.001) (Table 1).

### 3.2. Multiple Logistic Regression Models

Females exhibited significantly lower odds of seeking OSMI compared to males (OR = 0.51; 95% CI: [0.39–0.67]). Divorced status was closely correlated with OSMIS behavior compared to married or widowed (3.13 [1.92–5.17]). Respondents with secondary or higher levels of educational attainment were less likely to seek OSMI than those with only an elementary level of education (0.33 [0.21–0.50]). Unemployed people had higher odds of seeking OSMI (2.21 [1.64–2.99]) compared to pensioners. Living in a rural village was associated with greater odds of searching for OSMI (1.39 [1.12–1.93]) compared to residing in a suburb or a town/small city. Several income quintiles demonstrated a significant decrease in the odds of OSMIS behavior, with the second (0.47 [0.33–0.66]) and third quintiles (0.41 [0.26–0.64]) exhibiting the most pronounced reductions. Individuals reporting good self-rated health were associated with lower odds of seeking OSMI compared to those in poor self-perceived health (0.39 [0.23–0.68]). Unhappy people had significantly higher odds of searching OSMI compared to happy individuals (0.64 [0.49–0.84]). Individuals without chronic diseases had lower odds of searching for OSMI (0.46 [0.34–0.63]). Normal BMI was strongly connected to OSMIS behavior compared to overweight and obese (2.43 [1.64–4.07]). Catholic individuals had significantly higher odds of seeking OSMI than atheists or those of other religions (2.25 [1.56–3.46]). Mistrust in healthcare schemes was robustly linked to seeking OSMI (2.03 [1.33–3.11]. Individuals who reported difficulties accessing the healthcare system had higher odds of seeking OSMI (1.85 [1.39–2.46]). Respondents who had confidence in general online health information had significantly higher odds of seeking OSMI (2.02 [1.32–3.10]). Other societal and healthy lifestyle-related determinants consisting of subjective well-being, depression, attendance at religious services, smoking, alcohol use, and trust in physicians did not depict a significant association with OSMIS behavior (Table 2).

### 3.3. Sensitivity Analysis

Subpopulation analysis revealed that individuals from rural areas (2.81 [1.26–4.91]) with only elementary education (3.76 [2.21–6.05]), belonging to the lowest income quintile (2.11 [1.32–4.96]) exhibited higher odds of seeking OSMI. The robustness of the initial model was confirmed through bootstrap analysis with 1000 iterations. Key variables consistently remained significant, reinforcing both the stability of the associations and the model’s predictive accuracy for OSMIS behavior in the sample. The findings from bootstrapping and subpopulation analyses consistently supported the robustness of the original model.

## 4. Discussion

This study aims to scrutinize the demographic, healthy lifestyle, and socioeconomic factors affecting OSMIS behavior. Our findings suggest that OSMIS played a significant function in mitigating health inequalities contributing to developing mental conditions of socially excluded strata.

Men displayed significantly higher odds of seeking OSMI compared to women. This first empirical result is novel, as previous scientific investigations highlighted that women were overrepresented in online health information-seeking (OHIS) [27]. The sex discrepancy in OSMIS behavior may be explained by social role theory suggesting that men often face societal pressure to address issues independently, leading them to search for solutions online instead of asking for help in person [28,29]. Furman and Joseph [30] demonstrated that men may also be less inclined to openly treat mental health concerns, making the internet more convenient and anonymous for accessing OSMI. In contrast, traditional societal norms suggest that women, who are generally more open to face-to-face communication, may prefer discussing stress-related concerns in person with friends, family members, significant others, or health professionals, rather than seeking OSMI [31,32]; during the COVID-19 period in Hungary, men were more likely than women to seek OSMI. Overall, the sex discrepancies of OSMIS behavior must be scrutinized with qualitative and quantitative methods to understand their core traits.

Furthermore, our empirical data reveal that socially excluded classes, including unemployed individuals with low educational attainment and income, regularly sought out OSMI during the COVID-19 period [33]. It is particularly noteworthy that individuals with lower educational attainment also demonstrated a level of health literacy sufficient to enable them to seek OSMI. This process might have been facilitated by affordable internet access in Hungary. This represents a novel scientific finding since societal strata with higher prestige had predominantly utilized the advantages of digital health before the pandemic [34,35]. However, during the lockdowns, marginalized social classes were significantly exposed to various forms of stress and were often unable to cope with them adequately. Their skills for managing stressful situations in daily life were underdeveloped, and they lacked sufficient financial resources to access in-person mental health services. In this complex and challenging situation, the Hungarian individuals in poverty have commenced to search for free OSMI, offering practical information on how to respond to stress. Online stress management platforms were continuously improved in Hungary, offering practical information on coping techniques provided by psychiatrists, psychologists, and other health professionals. A tangible example of this is the website webbeteg.hu. Vulnerable social groups, however, also collected OSMI through social media platforms.

Our data also revealed that living in a smaller settlement was associated with OSMIS behavior. Drawing on previous findings, it can be asserted that individuals from lower socioeconomic strata in rural areas significantly sought OSMI during the pandemic [36]. Low income, limited employment opportunities, and persistent financial instability contributed to chronic uncertainty, exacerbating the prevalence of prolonged stress. Moreover, for rural populations, psychological counseling or therapy is often difficult to obtain due to geographic isolation, insufficient local resources, and financial constraints [37].

This multilayer situation has likely acted as a driving force of OSMIS behavior, particularly among individuals lacking alternative coping mechanisms. More precisely, OSMI emerges as a viable alternative, providing an accessible source of information on stress management that helps to compensate for the absence of locally available healthcare and social support systems.

Essentially, rural social classes with lower prestige actively sought OSMI and improved their coping skills, suggesting that health inequalities could be mitigated during the COVID-19 pandemic [38].

Working-age individuals (25–59-years group, or generation Y and X) significantly connected to OSMIS behavior. This phenomenon can be attributed to age-related stressors, which play a pivotal function in OSMIS behavior. Unemployed, middle-aged individuals often faced job search-related and child-rearing challenges and financial pressures, all of which exposed them to a greater number of chronic stressors [39]. Additionally, they often experienced more pronounced social role conflicts further exacerbating stress levels [40]. They were also more acutely aware of the long-term health consequences of stress (e.g., cardiovascular diseases, born-out syndrome, depression, and anxiety), which fostered greater health consciousness and led to more intentional and deliberate online information-seeking behaviors.

Social support serves a health-protective function against stress [41]. Individuals with robust social networks are better equipped to cope with stress due to the psychosocial resources provided by friends, family members, and significant others [42]. Social support enhances self-confidence and a sense of control, thereby increasing the effectiveness of stress management, and it fosters positive thinking, which facilitates coping. In brief, the buffering effect of social support mitigates the negative impacts of stress, anxiety, depression, and frustration. Consequently, divorced individuals lacking adequate social support were more likely to seek OSMI to preserve their mental well-being and receive informational support [43,44]. Divorce often results in heightened loneliness, social isolation, and financial instability, which amplify stress levels; this situation was further exacerbated by the COVID-19 lockdowns. However, OSMI offered a readily accessible and cost-free platform for stress management while providing secure anonymity and minimized exposure to social stigma. In summary, divorced individuals frequently engaged with OSMI to enhance their coping strategies and resilience.

Unhappy individuals with chronic disease and low self-perceived health status also had high odds of seeking OSMI. The complex interplay of these factors associated with chronic conditions is often linked to prolonged stress [45,46]. Living with chronic illness frequently entails persistent physical discomfort, uncertainty surrounding the progression of the disease, and the ongoing demands of managing medical treatments. These circumstances elevate stress levels, driving individuals to adopt effective coping strategies, such as OSMIS behavior. Furthermore, physical mobility limitations or the necessity of frequent medical appointments can render traditional in-person interventions or support groups less accessible. In such cases, the internet emerges as a convenient and readily available resource for stress management, providing access to OSMIS at any time, without requiring individuals to leave their homes. This accessibility makes OSMIS behavior particularly advantageous for those who face physical challenges in attending regular medical appointments.

However, not only individuals with low health and socioeconomic status searched for OSMI but people with normal BMI (body mass index). Individuals with a normal BMI tended to exhibit higher levels of health awareness, which drives their interest in topics related to lifestyle, health, and stress management. In contrast, people with obesity often demonstrated lower health consciousness, potentially limiting their motivation to seek such information. Furthermore, social standards and stigma also played a significant role: obese individuals frequently faced societal stigmatization, which could contribute to feelings of anxiety and inhibit their willingness to search online, particularly if they perceive stress or weight gain as a personal failing [47].

In general, for individuals with obesity, the experience of stress and approaches to coping are often distinct. Research has shown that obesity and stress may create a self-reinforcing cycle: obese individuals were more likely to engage in emotion-focused coping mechanisms (e.g., overeating) rather than problem-focused strategies, such as seeking online information on stress management. Those with a normal BMI, on the other hand, were more inclined toward proactive approaches to stress management.

Religious belief, including Catholicism, is a relevant protective factor of human mental health [48]. Practices such as prayer or church attendance often served as effective stress-relief mechanisms, fostering emotional and mental stability. However, when these traditional coping strategies prove inadequate in addressing the complexities of modern life, individuals may turn to supplementary resources, such as OSMI, to treat stressors. Religious beliefs were often linked to proactive health behaviors, including OSMIS behavior. Catholic teachings emphasize the importance of maintaining health as a divine gift, which may encourage Hungarian believers to not only engage in spiritual practices but also adopt modern health-preservation strategies, such as OSMIS behavior.

People who did not trust healthcare but had confidence in OSMI sought OSMI regularly. Individuals who distrusted the healthcare system often perceived that medical care failed to provide adequate medical support or that the healthcare system was unable to effectively address their serious concerns, including mental health and stress management issues. For those with limited access to healthcare services, whether due to geographic distance or financial constraints, the internet can serve as a valuable resource. In contrast, individuals with internet access and digital health literacy are more inclined to utilize online resources, particularly when trust in the healthcare system has diminished or when healthcare access is restricted. Sociological and digital health studies demonstrated that the internet democratizes access to OSMI, empowering individuals to make informed choices from available resources, especially in contexts where the healthcare system fails to provide adequate services.

Individuals who trusted OSMI generally possess basic levels of digital health literacy and were more adept at locating and interpreting relevant, credible sources [49]. Trust in online resources was often linked to the ability to critically assess the reliability and relevance of the information, making them more likely to engage in proactive information-seeking behavior, including the search for stress management techniques. Trust in OSMI represented an adaptive behavior, favoring modern, efficient, and convenient methods of stress management. In a nutshell, OSMI might provide a solution that is fast and convenient and it allows individuals to adopt modern, personalized forms of stress management.

Access to healthcare was significantly influenced by spatial and financial factors. For individuals living in rural areas or with lower incomes and prestige, private clinics represented an expensive and logistically challenging option, particularly for mental or psychological services. These services entail considerable time and financial investment, which many cannot afford. OSMIS behavior could fill this gap by enabling access to health information from anywhere, at any time. Turning to online resources can empower individuals to take greater agency in addressing their health challenges, fostering the role of the “empowered patient”.

These structural inequalities pushed them toward utilizing the internet as a cost-effective and readily accessible alternative; escalation of OSMIS behavior may be defined as a rational individual response to health inequalities. In other words, the search for online stress management techniques may thus reflect systemic limitations in the healthcare system, where a combination of welfare and market mechanisms might leave marginalized groups at a disadvantage.

## 5. Conclusions

This paper provided insights into the societal, demographic, and health-related lifestyle determinants of OSMIS behavior. The results revealed that marginalized social groups utilized digital health devices to gather OSMI as a means of coping with daily stress. Specifically, divorced, unhappy men from the X and Y generations, those experiencing chronic illnesses, dissatisfaction, or limited access to healthcare were more likely to seek OSMI. In addition, a unique combination of trust in digital health resources and distrust in traditional healthcare systems played a pivotal function in fostering proactive OSMIS behavior. The novelty of this study lies in underscoring the fact that vulnerable social classes had limited access to digital health before COVID-19. However, from 2021, marginalized groups began to apply these technologies to mitigate the multilayered impact of stress, potentially reducing health disparities.

In conclusion, during the COVID-19 pandemic, the concept of “techquity” emerged as a key factor in Hungary. Digital tools have facilitated access to information, techniques, and methods for stress management among vulnerable social groups, thereby contributing to the reduction in inequalities. OSMIS behavior reflects a rational adaptation to structural health inequalities and shifting social dynamics, serving as a bridge for marginalized populations in navigating mental health challenges. In this respect, these results may be considered in the development of digital health devices, health policy decision-making, and clinical practice in order to prevail digital inclusion in healthcare [50]. More precisely, health applications for stress management may be developed and monitored by health professionals and should be promoted and made accessible to vulnerable social strata. This complex process can be managed and financed by health policy actors.

## 6. Strength and Limitations

An applied standardized questionnaire and a large, nationally representative ISSP 2021 dataset ensured that results could be generalized to accurately reflect the attributes of the Hungarian population. However, data were collected during the COVID-19 lockdown, a period of increased internet usage, which might alter societal patterns of OSMIS behavior. Moreover, the ISSP 2021 dataset was composed of self-reported data, which might bias the results; therefore, empirical findings should be interpreted with caution. Finally, we elaborated a cross-sectional dataset, which did not allow us to determine causal or temporal correlations.

## Figures and Tables

**Table 1 ijerph-22-00473-t001:** Demographic, socioeconomic, and healthy lifestyle characteristics of study participants in weighted proportions (%) and unweighted numbers (n).

Variable	Category	Non-Users of OSMI (n,%)	Users of OSMI (n,%)	Total (n,%)	*p*-Value
Sex	Male	290 (46.3)	114 (30.9)	404 (40.6)	**<0.001**
	Female	336 (53.7)	255 (69.1)	591 (59.39)	
Age group	Gen Z (18–24 y)	32 (5.1)	14 (3.8)	46 (4.62)	**<0.001**
	Gen Y (25–43 y)	141 (22.5)	139 (37.7)	280 (28.14)	
	Gen X (44–59 y)	218 (34.8)	133 (36)	351 (35.27)	
	Older adult (60 y-)	235 (37.5)	83 (22.5)	318 (31.95)	
Marital status	Single	111 (17.7)	80 (21.7)	191 (19.23)	**<0.001**
	Married	327 (52.2)	215 (58.3)	542 (54.58)	
	Divorced	85 (13.6)	52 (14.1)	137 (13.79)	
	Widowed	101 (16.1)	22 (6)	123 (12.4)	
Educational attainment	Primary	132 (21.1)	30 (8.1)	162 (16.28)	**<0.001**
	Secondary	422 (67.4)	285 (77.2)	707 (71.05)	
	Higher	72 (11.5)	54 (14.6)	126 (12.66)	
Place of residence	Big city	219 (35)	136 (36.9)	355 (35.67)	**<0.001**
	Suburb	16 (2.6)	31 (8.4)	47 (4.72)	
	Town/small city	194 (31)	113 (30.6)	307 (30.85)	
	Rural village	197 (31.5)	89 (24.1)	286 (28.74)	
Subjective well-being	Poor	61 (9.9)	38 (10.4)	99 (10.09)	**<0.001**
	Lower mid class	180 (29.2)	96 (26.3)	276 (28.13)	
	Middle class	187 (30.4)	73 (20)	260 (26.5)	
	Upper mid class	158 (25.6)	127 (34.8)	285 (29.05)	
	Affluent	30 (4.9)	31 (8.5)	61 (6.21)	
Income quintiles	First	211 (47.4)	83 (29.9)	294 (40.66)	**<0.001**
	Second	145 (32.6)	120 (43.2)	265 (36.65)	
	Third	62 (13.9)	58 (20.9)	120 (16.59)	
	Fourth	5 (1.1)	6 (2.2)	11 (1.52)	
	Fifth	22 (4.9)	11 (4)	33 (4.56)	
Smoking	Smoker	171 (27.3)	106 (28.7)	277 (27.86)	0.728
	Non-smoker	455 (72.7)	262 (71)	717 (72.13)	
Alcohol use	Drinker	21 (3.4)	20 (5.4)	41 (4.12)	**<0.001**
	Non-drinker	604 (96.5)	349 (94.6)	953 (95.87)	
Had depression	Yes	161 (25.7)	109 (29.5)	270 (27.16)	0.144
	No	464 (74.1)	260 (70.5)	724 (72.83)	
Happiness	Not happy	270 (42.2)	118 (32)	388 (38.5)	**<0.001**
	Happy	369 (57.8)	250 (68)	619 (61.5)	
Self-perceived health	Not good	69 (10.7)	17 (4.6)	86 (8.5)	**<0.001**
	Good	570 (89.3)	352 (95.4)	922 (91.5)	
Had chronic disease	Yes	201 (32.1)	67 (18.2)	268 (26.96)	**<0.001**
	No	425 (67.9)	301 (81.8)	726 (73.03)	
Employment	Employed	366 (60)	271 (75.6)	637 (65.8)	**<0.001**
	Unemployed	233 (38.2)	79 (22)	312 (32.2)	
	Pensioner	10 (1.7)	8 (2.3)	18 (2)	
BMI	Overweight and obese	150 (24.1)	93 (25.6)	243 (24.7)	**<0.04**
	Normal	471 (75.9)	269 (74.4)	740 (75.3)	
Religion	Other religions	139 (22.4)	117 (32.5)	256 (26.2)	**<0.001**
	Atheist	96 (15.5)	71 (19.7)	167 (17)	
	Catholic	383 (62.1)	171 (47.8)	554 (56.8)	
Attendance at religious service	Several times in a month	63 (10.1)	28 (7.7)	91 (9.2)	0.09
	Several times in a year	295 (47.4)	199 (54.8)	494 (50.1)	
	Never	264 (42.5)	136 (37.5)	400 (40.7)	
Trust in healthcare	Yes	536 (84.1)	335 (91.5)	871 (86.8)	**<0.001**
	No	101 (15.9)	31 (8.5)	132 (13.2)	
Trust in doctors	Yes	616 (97.4)	351 (95.9)	967 (96.8)	0.16
	No	16 (2.6)	15 (4.1)	31 (3.2)	
Easy access to healthcare	Yes	379 (60.8)	268 (74.2)	647 (65.7)	**<0.001**
	No	244 (39.2)	93 (25.8)	337 (34.3)	
Trust in online health information	Yes	418 (83.43)	332 (90.95)	750 (86.6)	**<0.001**
	No	83 (16.56)	33 (9)	116 (13.39)	

Bold values indicate statistical significance (*p* < 0.05) according to Pearson’s chi-squared test.

**Table 2 ijerph-22-00473-t002:** Weighted multiple logistic regression analysis of factors impacting OSMIS behavior prevalence.

Characteristics		Users of Online Stress Management Information	
		OR (95% CI)	*p*-Value
Sex	Male		
	Female	**0.51 [0.39–0.67]**	**<0.001**
Age group	Gen Z (18–24 y)		
	Gen Y (25–43 y)	**3.42 [1.51–4.81]**	**0.03**
	Gen X (44–59 y)	**2.01 [1.35–3.34]**	**<0.001**
	Older adults (60 y-)	1.18 [0.60–2.31]	0.61
Marital status	Single		
	Married	1.64 [0.31–8.54]	0.55
	Divorced	**3.13 [1.92–5.17]**	**<0.001**
	Widowed	0.94 [0.67–1.31]	0.74
Educational attainment	Primary		
	Secondary	**0.33 [0.21–0.50]**	**<0.001**
	Higher	**0.31 [0.18–0.53]**	**<0.001**
Employment	Employed		
	Unemployed	**2.21 [1.64–2.99]**	**<0.001**
	Pensioner	0.91 [0.35–2.34]	0.85
Place of residence	Big city		
	Suburb	0.31 [0.16–0.60]	<0.24
	Town/small city	1.07 [0.78–1.47]	0.54
	Rural village	**1.39 [1.12–1.93]**	**<0.001**
Subjective well-being	Poor		
	Lower mid class	1.11 [0.69–1.79]	0.63
	Middle class	1.53 [0.94–2.49]	0.08
	Upper mid class	0.74 [0.47–1.19]	0.22
	Affluent	0.57 [0.30–1.09]	0.09
Income quintiles	First		
	Second	**0.47 [0.33–0.66]**	**<0.001**
	Third	**0.41 [0.26–0.64]**	**<0.001**
	Fourth	0.32 [0.09–1.08]	0.56
	Fifth	0.77 [0.35–1.66]	0.68
Smoking	Non-smoker		
	Smoker	1.04 [0.83–1.31]	0.70
Alcohol use	Non-drinker		
	Drinker	0.95 [0.72–1.24]	0.70
Had depression	Yes		
	No	1.04 [0.83–1.31]	0.67
Happiness	Not happy		
	Happy	**0.64 [0.49–0.84]**	**<0.001**
Self-perceived health	Not good		
	Good	**0.39 [0.23–0.68]**	**<0.001**
Had chronic disease	Yes		
	No	**0.46 [0.34–0.63]**	**<0.001**
BMI	Overweight and obese		
	Normal	**2.25 [1.56–3.46]**	**<0.001**
Religion	Other religion		
	Atheist	1.13 [0.76–1.68]	0.51
	Catholic	**1.88 [1.39–2.55]**	**<0.001**
Attendance at religious services	Several times in a month		
	Several times in a year	0.52 [0.20–1.35]	0.48
	Never	0.68 [0.26–1.77]	0.43
Trust in healthcare	Yes		
	No	**2.03 [1.33–3.11]**	**<0.001**
Trust in doctors	Yes		
	No	0.60 [0.29–1.24]	0.77
Easy access to healthcare	Yes		
	No	**1.85 [1.39–2.46]**	**<0.001**
Trust in online health information	No		
	Yes	**2.02 [1.32–3.10]**	**<0.001**

Bold values depict statistical significance (*p* < 0.05). The ORs are adjusted for other variables in the model.

## Data Availability

The data collected for this study might be available on request from the corresponding author.

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
