# Peer review of "Patterns of Online Stress Management Information-Seeking Behavior in Hungary"

_ijerph, 2025, doi:10.3390/ijerph22040473_

Round 1
Reviewer 1 Report
Comments and Suggestions for Authors
Thank you for inviting me to review the manuscript "Patterns of Online Stress Management Information-Seeking Behavior in Hungary." The authors provide an interesting study on the societal, demographic, and health-related determinants of online stress management information-seeking (OSMIS) behavior in Hungary. Overall, the manuscript is well-written and well-organized, but several aspects require the authors' attention to improve its quality:
-The abstract effectively summarizes the study, but it could benefit from including specific numerical results (e.g., odds ratios or confidence intervals) to highlight key findings.
-Section 1 Introduction. The following statement lacks clarity: "While numerous empirical studies have examined various patterns of online health information-seeking, the specifics of online stress management information-seeking (OSMIS) behavior remain underexplored [14, 15, 16]. This paper addresses this gap." It would be helpful for the authors to define the gap being addressed clearly. Justify this based on the state of the art and explicitly articulate the scientific contribution of the study. Emphasizing how this work uniquely contributes to the existing literature would strengthen the manuscript.
-Section 2 Methodology. While the methodology is described in detail, additional information about the selection criteria for the dataset variables would improve reproducibility and clarity.
-Section 4 Discussion. Emphasize the implications of the results obtained. Specifically, how can these findings be applied in practice or influence future research, policy, or interventions?
I hope these suggestions will assist the authors in refining their manuscript.
Author Response
Thank you for inviting me to review the manuscript "Patterns of Online Stress Management Information-Seeking Behavior in Hungary." The authors provide an interesting study on the societal, demographic, and health-related determinants of online stress management information-seeking (OSMIS) behavior in Hungary. Overall, the manuscript is well-written and well-organized, but several aspects require the authors' attention to improve its quality:
Comment: The abstract effectively summarizes the study, but it could benefit from including specific numerical results (e.g., odds ratios or confidence intervals) to highlight key findings.
Response: Thank you for pointing this out. Please see lines 17-20 in the corrected manuscript for our answer.
Comment: Section 1 Introduction. The following statement lacks clarity: "While numerous empirical studies have examined various patterns of online health information-seeking, the specifics of online stress management information-seeking (OSMIS) behavior remain underexplored [14, 15, 16]. This paper addresses this gap." It would be helpful for the authors to define the gap being addressed clearly. Justify this based on the state of the art and explicitly articulate the scientific contribution of the study. Emphasizing how this work uniquely contributes to the existing literature would strengthen the manuscript.
Response: Thank you for pointing this out. Please see lines 55-59 in the corrected manuscript for our answer.
Comment: Section 2 Methodology. While the methodology is described in detail, additional information about the selection criteria for the dataset variables would improve reproducibility and clarity.
Response: Thank you for pointing this out. Please see lines 87-111 in the corrected manuscript for our answer.
Comment: Section 4 Discussion. Emphasize the implications of the results obtained. Specifically, how can these findings be applied in practice or influence future research, policy, or interventions?
Response: Thank you for pointing this out. Please see lines 386-389, and 397-400 in the corrected manuscript for our answer.
Reviewer 2 Report
Comments and Suggestions for Authors
Comments are included in the document.

Author Response
Patterns of Online Stress Management Information-Seeking Behavior in Hungary
1.MATERIALS AND METHODS
Comment: Kindly indicate the approach used before talking about the designs.
Response: Thank you for pointing this out. Please see lines 76-140 in the corrected manuscript for our answer.
Comment: The sub-topic of 2.2. the sampling process should include the sampling method, sample size, and population. 2.3. should be data collection. Under that topic, it should be a data collection instrument and data collection process.
Response: Thank you for pointing this out. Please see lines 87-140 in the corrected manuscript for our answer.
Comment: Regarding the questionnaire, indicate whether it was self-developed or adopted. If it was adopted, was the permission sought?
Response: Thank you for pointing this out. Please see lines 78-81 in the corrected manuscript for our answer.
Comment: Indicate the sub-sections included in the questionnaire.
Response: Thank you for pointing this out. Please see lines 88-111 in the corrected manuscript for our answer. Moreover, Appendix 1 includes the Hungarian and English questionnaire.
Comment: Kindly put the information on the characteristics of respondents in a table for clear understanding.
Response: Thank you for pointing this out. Please see lines 163-187 in the corrected manuscript for our answer.
2.RESULTS
Comment: Indicate the reasons for choosing logistic regression instead of linear regression.
The sub-topics in 3.1. can be the respondents' characteristics. Under it, you write the demographic data, socioeconomic/employment status, and healthy lifestyle.
Response: Multiple logistic regression can be applied when the dependent variable is dichotomous (binary; seek or not seek OSMI). This is the main reason we used logistic regression analysis. Other scientific papers also utilized this methodology (see reference literature in this chapter).
3.DISCUSSIONS
Comment: Conducted studies must support the information between lines 309 and 329.
Response: Thank you for pointing this out. Please see line 336 in the corrected manuscript for our answer.
Comment: The referencing style should be the same in the list and the intext information. In the intext, Vancouver was used, and another one was on the list.
Response: Thank you for pointing this out. Please see lines 163-187 in the corrected manuscript for our answer.
Response: It was corrected in the final version of the manuscript.
Reviewer 3 Report
Comments and Suggestions for Authors
The manuscript entitled “Patterns of Online Stress Management Information-Seeking Behaviour in Hungary” was interesting. The authors aimed to examine the societal, demographic, and health-related determinants of online stress management information-seeking (OSMIS) behaviour in Hungary. However, there are some issues which need further attentions:
- Keywords should be selected based on the MeSH terms.
- In the introduction, the authors said “….the specifics of online stress management information-seeking (OSMIS) behaviour remain underexplored”. It is not clear what they mean by “specifics”.
- The time of the study is not clear.
- It is not clear how the participants were recruited and who they were.
- More information about the standardized questionnaire is required. How was it developed?
- A copy of the questionnaire can be provided as an Appendix.
- Please use the term “sex” instead of “gender”.
- It is not clear whether this study was conducted during Covid-19 or post-covid-19. The date of data collection influences the interpretation of the results, and even the necessity of mentioning or not mentioning Covid-19.
- While the results present current status of OSMIS in Hungary, I was not convinced about the novelty of the research.
Author Response
The manuscript entitled “Patterns of Online Stress Management Information-Seeking Behaviour in Hungary” was interesting. The authors aimed to examine the societal, demographic, and health-related determinants of online stress management information-seeking (OSMIS) behaviour in Hungary. However, there are some issues which need further attentions:
Comment: In the introduction, the authors said “….the specifics of online stress management information-seeking (OSMIS) behaviour remain underexplored”. It is not clear what they mean by “specifics”.
Response: Thank you for pointing this out. Please see lines 55-59 in the corrected manuscript for our answer.
Comment: The time of the study is not clear.
Response: Thank you for pointing this out. Please see lines 78-81 in the corrected manuscript for our answer.
Comment: It is not clear how the participants were recruited and who they were.
Response: Thank you for pointing this out. Please see lines 102-111 in the corrected manuscript for our answer.
Comment: More information about the standardized questionnaire is required. How was it developed?
Response: Thank you for pointing this out. Please see lines 87-102 in the corrected manuscript for our answer.
Comment: A copy of the questionnaire can be provided as an Appendix.
Response: Thank you for pointing this out. Please see line 97 in the corrected manuscript for our answer.
Comment: Please use the term “sex” instead of “gender”.
Response: Thank you, it was corrected.
Comment: It is not clear whether this study was conducted during Covid-19 or post-covid-19. The date of data collection influences the interpretation of the results, and even the necessity of mentioning or not mentioning Covid-19.
Response: Thank you for pointing this out. Please see lines 79-81 in the corrected manuscript for our answer.
Comment: While the results present the current status of OSMIS in Hungary, I was not convinced about the novelty of the research.
Response: Thank you for pointing this out. Please see lines 250-267, 386-389 in the corrected manuscript for our answer.
Reviewer 4 Report
Comments and Suggestions for Authors
Dear authors, first of all, thank you for the opportunity to review your manuscript.
I would like to inform you that I have read your manuscript carefully and I believe that the following improvements should be made:
Introduction.
I think it is well developed, however, I think it is prudent to make the following adjustments:
- To give it greater forcefulness, it is necessary to record the prevalence of mental health in the Hungarian population.
- Although I understand that the OSMIS is relevant, I believe that the authors should record a slightly more critical position on the matter. What I mean is that they should make it clear that not all the information available on the Internet has scientific evidence to reduce stress.
- I also believe that the OSMIS should be operationalized in a better way.
Methodology
- Line 92: Please incorporate statistical calculations in the methodology to indicate that the survey was representative for the Hungarian context. How do you arrive at this assertion?
- In the questionnaire section, please include some sample questions from this instrument.
- What is the question or questions that give rise to the OSMIS variable?
Data analysis:
Given that multiple sociodemographic, economic and lifestyle variables are used, it would be advisable to include a collinearity analysis using the variance inflation factor (VIF) and correlation matrix.
Results
Table 1 is very well presented, clear and in order.
Table 2 is very well done, very clear.
I think that the results section has been very well done, the summary at the end of the section gives an account of the most relevant variables to analyze in the discussion.
Discussion.
The discussion is very well done, however it would be good to move forward with these points:
- Favor recording how Hungarian cultural or social variables could be interfering in your results. The result of the gender established in the results is interesting.
- A paragraph should be included in which the results of the present study are discussed with other results in other articles.
- Include a section where the results obtained are compared with previous studies in other countries or regions.
- There is an interesting finding that people with secondary or higher education are less likely to search for OSMIS, this seems interesting to me to discuss in more depth.
- It is striking that it is mentioned that the COVID-19 pandemic increased the OSMIS, but this is not discussed, it is necessary to go deeper into this, it seems relevant to me especially due to the date of data collection.
Limitations
The following limitations should be included:
- The data was obtained by self-report... could that not affect?
- The OSMIS is analyzed, however, what is the quality of that information? Is it validated through strategies with scientific validation?
- The periodicity of the OSMIS could not be established, will it be the same to consult once or every day?
- There are also no deeper psychological variables such as personality traits, nor do we know if these people have any other pathology, which could, for example, lead to greater compulsion.
Author Response
Dear authors, first of all, thank you for the opportunity to review your manuscript.
I would like to inform you that I have read your manuscript carefully and I believe that the following improvements should be made:
1.Introduction.
I think it is well developed, however, I think it is prudent to make the following adjustments:
Comment: To give it greater forcefulness, it is necessary to record the prevalence of mental health in the Hungarian population.
Response: Thank you for pointing this out. Please see lines 29-35 in the corrected manuscript for our answer.
Comment: Although I understand that the OSMIS is relevant, I believe that the authors should record a slightly more critical position on the matter. What I mean is that they should make it clear that not all the information available on the Internet has scientific evidence to reduce stress.
Response: Thank you for pointing this out. The chapter discussion was refined utilizing this approach; please see lines 231-377 in the corrected manuscript for our answer.
Comment: I also believe that the OSMIS should be operationalized in a better way.
Response: Thank you for pointing this out. Please see lines 61-63 in the corrected manuscript for our answer.
Methodology
Comment: Line 92: Please incorporate statistical calculations in the methodology to indicate that the survey was representative for the Hungarian context. How do you arrive at this assertion?
Response: Thank you for pointing this out. Please see lines 102-106 in the corrected manuscript for our answer.
Comment: In the questionnaire section, please include some sample questions from this instrument.
Response: Thank you for pointing this out. Please see lines 114-121 and 97 in the corrected manuscript for our answer.
Comment: What is the question or questions that give rise to the OSMIS variable?
Response: Thank you for pointing this out. Please see lines 114-121 in the corrected manuscript for our answer.
Discussion.
The discussion is very well done, however it would be good to move forward with these points:
Comment: Favor recording how Hungarian cultural or social variables could be interfering in your results. The result of the gender established in the results is interesting.
- A paragraph should be included in which the results of the present study are discussed with other results in other articles.
- Include a section where the results obtained are compared with previous studies in other countries or regions.
Response: As we indicated, we did not find scientific papers which analyze the same aspect of online health information seeking.
Comment: There is an interesting finding that people with secondary or higher education are less likely to search for OSMIS, this seems interesting to me to discuss in more depth.
Response: Thank you for pointing this out. Please see lines 250-267 in the corrected manuscript for our answer.
Comment: It is striking that it is mentioned that the COVID-19 pandemic increased the OSMIS, but this is not discussed, it is necessary to go deeper into this, it seems relevant to me especially due to the date of data collection.
Response: Thank you for pointing this out. Unfortunately, we do not have sufficient statistical data on this phenomenon. Please see lines 404-405 in the corrected manuscript for our answer.
Limitations
Comment: The following limitations should be included: The data was obtained by self-report... could that not affect?
Response: Thank you for pointing this out. Please see lines 406-407 in the corrected manuscript for our answer.
Comment: The OSMIS is analyzed, however, what is the quality of that information? Is it validated through strategies with scientific validation?
Response: The ISSP 2021 dataset does not provide any units of information on the quality of OSMI. The paper scrutinizes only the societal patterns of online stress management information-seeking.
Comment: The periodicity of the OSMIS could not be established, will it be the same to consult once or every day?
Response: Thank you for pointing this out. Please see lines 114-121 in the corrected manuscript for our answer.
Comment: There are also no deeper psychological variables such as personality traits, nor do we know if these people have any other pathology, which could, for example, lead to greater compulsion.
Response: The dataset does not include these variables.
Round 2
Reviewer 3 Report
Comments and Suggestions for Authors
Thank you very much for your email. I appreciate the authors' time and efforts to revise the manuscript. It has been improved significantly and can be accepted and published in the journal.